# The Solubility of Ethyl Candesartan in Mono Solvents and Investigation of Intermolecular Interactions

Cunbin Du 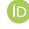





School of Pharmaceutical Sciences, TaiZhou University, Taizhou 318000, China; cbd@tzc.edu.cn

**Abstract:** In this work, the experimental solubility of ethyl candesartan in the selected solvents within the temperature ranging from 278.15 to 318.15 K was studied. It can be easily found that the solubility of ethyl candesartan increases with the rising temperature in all solvents. The maximum solubility value was obtained in *N,N*-dimethylformamide (DMF, $7.91 \times 10^{-2}$), followed by cyclohexanone ($2.810 \times 10^{-2}$), 1,4-dioxanone ($2.69 \times 10^{-2}$), acetone ($7.04 \times 10^{-3}$), ethyl acetate ($4.20 \times 10^{-3}$), *n*-propanol ($3.69 \times 10^{-3}$), isobutanol ($3.38 \times 10^{-3}$), methanol ($3.17 \times 10^{-3}$), *n*-butanol ($3.03 \times 10^{-3}$), ethanol ($2.83 \times 10^{-3}$), isopropanol ($2.69 \times 10^{-3}$), and acetonitrile ($1.15 \times 10^{-2}$) at the temperature of 318.15 K. Similar results of solubility sequence from large to small were also obtained in other temperatures. The X-ray diffraction analysis illustrates that the crystalline forms of all samples were consistent, and no crystalline transformation occurred during the dissolution process. In aprotic solvents, except for individual solvents, the solubility data decreases with the decreasing values of hydrogen bond basicity ($\beta$) and dipolarity/polarizability ($\pi^*$). The largest average relative deviation (*ARD*) data in the modified Apelblat equation is 1.9% and observed in isopropanol; the maximum data in $\lambda h$ equation is 4.3% and found in *n*-butanol. The results of statistical analysis show that the modified Apelblat equation is the more suitable correlation of experimental data for ethyl candesartan in selected mono solvents at all investigated temperatures. In addition, different parameters were used to quantify the solute–solvent interactions that occurred in the dissolution process including Abraham solvation parameters ($AP_i$), Hansen solubility parameters ($HP_i$), and Catalan parameters ($CP_i$).

**Keywords:** ethyl candesartan; solubility; model correlation; intermolecular interactions

## 1. Introduction

Candesartan (CNS) is a highly effective, long-acting, and selective angiotensin II type 1 receptor antagonist [1]. Candesartan cilexetil is a prodrug of candesartan, which can be completely hydrolyzed in the gastrointestinal tract and transformed into candesartan with antihypertensive activity [2]. At present, according to the different key intermediates in the synthesis process, there are many literature reports on the synthesis methods of candesartan cilexetil [3–7]. Among them, candesartan cilexetil is widely used by using 2-tert-butoxycarbonylamino-3-nitrobenzoate ethyl ester as a raw material through nine steps. However, in the process of triphenyl removal, due to the existence of multiple chemical sensitive groups in trityl candesartan cilexetil, impurities such as ethyl candesartan and incomplete materials will be produced. The purity of the product is low, and it needs to be purified many times to obtain candesartan ester with high purity [3]. EP 0720982 discloses a method for preparing candesartan cilexetil by deprotection of triphenylmethane in the presence of methanol and hydrochloric acid. The disadvantage of this process is that the yield is very low, and the product needs to be purified by chromatography [4].

In the process development and research of candesartan cilexetil, ethyl candesartan (Figure 1, chemical formula, $C_{26}H_{24}N_6O_3$, CAS No. 139481-58-6) is one of the important impurities in the preparation, and there is little literature on it. As we all know, the impurity profile of active pharmaceutical ingredients (API) and the evaluation of their toxic effects are necessary steps in the development of effective drugs, which is very important for

medical safety. Therefore, the basic knowledge required for any drug is its impurities and possible degradation products [8]. Solvent crystallization is a common method for the separation and purification step during the production process. The solubility of impurities in different solvents plays an important role for understanding the phase equilibrium in the development of the crystallization process [9–12]. Moreover, the solubility of a substance is determined by both the solid state (crystal lattice energy) and the interaction with the solvent (solvation) [9–12].

**Figure 1.** The chemical structure of ethyl candesartan.

Therefore, the proposed research work was performed to study the solubilization behavior of ethyl candesartan in some different pure solvents. The solubility data was correlated by a modified Apelblat equation and $\lambda h$ equation. The crystal form before and after dissolution was characterized using X-ray powder diffractometer. Moreover, the solubilization behavior was discussed by using the solvent properties. The solute–solvent interactions that occurred in the dissolution process were quantified by Abraham solvation parameters ($AP_i$), Hansen solubility parameters ($HP_i$), and Catalan parameters ($CP_i$). The physicochemical data obtained would be useful in purification, recrystallization, and formulation development of ethyl candesartan in pharmaceutical industries.

## 2. Experimental Section

### 2.1. Materials and Apparatus

Raw ethyl candesartan was recrystallized by ethanol; the final mass fraction purity was 0.992 (determined by the High Performance Liquid Chromatograph, HPLC, Agilent 1260, Beijing, China), provided by Zhejiang Junfeng Technology Co., Ltd., Taizhou, China. During the experiment, all pure organic solvents with analytical grade were purchased from Sinopharm Chemical Reagent Co., Ltd., Beijing, China, and used without any additional purification. The purity of the solvent was provided by the supplier. The detailed information is listed in Table 1.

**Table 1.** Source and purity of the materials used in this work.

| Chemicals | CAS Number | Molar Mass g·mol$^{-1}$ | Source | Mass Fraction Purity | Analysis Method |
|---|---|---|---|---|---|
| Ethyl Candesartan | 139481-58-6 | 468.51 | Zhejiang Junfeng Technology Co., Ltd. China | 0.992 | HPLC [b] |
| Methanol | 67-56-1 | 32.04 | | 0.995 [a] | |
| Ethanol | 64-17-5 | 46.07 | | 0.996 [a] | |
| *n*-Propanol | 71-23-8 | 60.10 | | 0.995 [a] | |
| Isopropanol | 67-63-0 | 60.10 | | 0.996 [a] | |
| *n*-Butanol | 71-36-3 | 74.12 | | 0.995 [a] | |
| Isobutanol | 78-83-1 | 74.12 | Sinopharm Chemical Reagent Co., Ltd., China | 0.996 [a] | None |
| Acetonitrile | 75-05-8 | 41.05 | | 0.996 [a] | |
| Ethyl Acetate | 141-78-6 | 88.11 | | 0.995 [a] | |
| DMF | 68-12-2 | 73.09 | | 0.995 [a] | |
| Acetone | 67-64-1 | 58.08 | | 0.996 [a] | |
| Cyclohexanone | 108-94-1 | 98.14 | | 0.995 [a] | |
| 1,4-Dioxane | 123-91-1 | 88.11 | | 0.996 [a] | |

[a] the purity was obtained from chemical reagent Co., Ltd., [b] determined by HPLC.

### 2.2. X-ray Diffraction Analysis

The crystals from raw ethyl candesartan and the recovered equilibrated samples from each solvent were analyzed by using X-ray powder diffractometer with an X-ray generator of Cu-Ka radiation (1.5405 Å). The experimental tube voltage and current were 40 kV and 30 mA. The data collection was performed at 2θ of 5–60° in steps of 0.02°.

### 2.3. Measurement Experiment

In this work, the isothermal saturation method [13–19] was used to determine the solubility data of ethyl candesartan in each solvent in the temperature ranging from 278.15 K to 318.15 K under atmospheric pressure. For solubility measurement, a jacketed glass vessel with a magnetic stirrer was used, and the temperature maintained by a thermostatic bath with an accuracy of 0.01 K.

Excess raw ethyl candesartan and 30 mL of solvent were added into the jacketed glass vessel. The actual temperature in solution was displayed by a mercury glass micro thermometer. A magnetic stirrer was used to mix continuously for 24 h to achieve phase equilibrium state. Then, the magnetic stirrer was stopped, and solution was settled for 2 h before sampling. Equilibrium liquor with the amount of 2 mL was taken out using a syringe attached with a 0.2 μm pore filter and transferred into a 25 mL pre-weighted flask covered with a rubber stopper, then weighed again by the analytical balance. After that, the concentration of ethyl candesartan was determined by HPLC. Each analysis was repeated three times at all temperatures. The mobile phase was methanol/water = (2:1) at a flow rate of 1.0 mL·min$^{-1}$. A reverse phase column LP-C18 (250 mm × 4.6 mm), with a column temperature of 303.15 K, and a UV detector, with the wavelength of 270 nm, were applied.

## 3. Results and Discussion

### 3.1. X-ray Diffraction Analysis

Figure 2 presents the XRD patterns of the crystals from raw ethyl candesartan and the recovered equilibrated samples from each solvent. The peaks in the raw material match well with the recovered equilibrated samples, which illustrates that the crystalline forms of all samples were consistent, and no crystalline transformation occurred during the dissolution process.

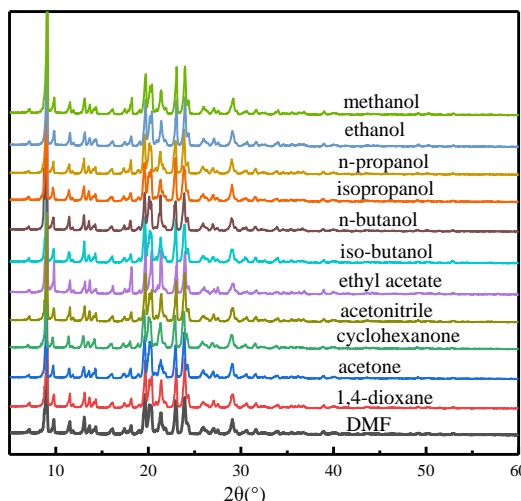

**Figure 2.** The XRD curves of ethyl candesartan recrystallized from each solvent.

### 3.2. Experimental Solubility Data

Through a search of related literature, we have found that previous studies mainly focused on the synthesis of ethyl candesartan. The quantitative solubility values in any of the investigated organic solvents are not reported yet. Therefore, the experimental solubility of ethyl candesartan in the selected solvents within the temperature range of

278.15 to 318.15 K was studied, and the solubility data together with the calculated values on the basis of correlation equation were tabulated in Table 2.

**Table 2.** Experimental and calculated solubility in mole fraction of ethyl candesartan in solvent at the temperature range of $T$ = (278.15 to 318.15) K under 101.1 kPa [a].

| $T$/K | \multicolumn{12}{c}{Solvent} |
|---|---|---|---|---|---|---|---|---|---|---|---|---|

| $T$/K | $x^{exp}$ | $x^{Ap}$ | $X^{\lambda h}$ | $x^{exp}$ | $x^{Ap}$ | $x^{\lambda h}$ | $x^{exp}$ | $x^{Ap}$ | $x^{\lambda h}$ | $x^{exp}$ | $x^{Ap}$ | $x^{\lambda h}$ |
|---|---|---|---|---|---|---|---|---|---|---|---|---|
| | **Methanol** | | | **Ethanol** | | | **n-Propanol** | | | **Isopropanol** | | |
| 278.15 | $3.81 \times 10^{-4}$ | $3.66 \times 10^{-4}$ | $3.98 \times 10^{-4}$ | $3.25 \times 10^{-4}$ | $3.28 \times 10^{-4}$ | $3.44 \times 10^{-4}$ | $5.17 \times 10^{-4}$ | $5.06 \times 10^{-4}$ | $5.27 \times 10^{-4}$ | $3.10 \times 10^{-4}$ | $2.87 \times 10^{-4}$ | $3.14 \times 10^{-4}$ |
| 283.15 | $5.24 \times 10^{-4}$ | $5.23 \times 10^{-4}$ | $5.37 \times 10^{-4}$ | $4.64 \times 10^{-4}$ | $4.59 \times 10^{-4}$ | $4.64 \times 10^{-4}$ | $6.67 \times 10^{-4}$ | $6.89 \times 10^{-4}$ | $6.95 \times 10^{-4}$ | $4.18 \times 10^{-4}$ | $4.14 \times 10^{-4}$ | $4.28 \times 10^{-4}$ |
| 288.15 | $7.04 \times 10^{-4}$ | $7.28 \times 10^{-4}$ | $7.16 \times 10^{-4}$ | $6.24 \times 10^{-4}$ | $6.29 \times 10^{-4}$ | $6.20 \times 10^{-4}$ | $9.04 \times 10^{-4}$ | $9.19 \times 10^{-4}$ | $9.07 \times 10^{-4}$ | $5.59 \times 10^{-4}$ | $5.81 \times 10^{-4}$ | $5.76 \times 10^{-4}$ |
| 293.15 | $9.80 \times 10^{-4}$ | $9.85 \times 10^{-4}$ | $9.46 \times 10^{-4}$ | $8.36 \times 10^{-4}$ | $8.44 \times 10^{-4}$ | $8.21 \times 10^{-4}$ | $1.22 \times 10^{-3}$ | $1.20 \times 10^{-3}$ | $1.17 \times 10^{-3}$ | $7.76 \times 10^{-4}$ | $7.94 \times 10^{-4}$ | $7.68 \times 10^{-4}$ |
| 298.15 | $1.32 \times 10^{-3}$ | $1.30 \times 10^{-3}$ | $1.24 \times 10^{-3}$ | $1.12 \times 10^{-3}$ | $1.11 \times 10^{-3}$ | $1.08 \times 10^{-3}$ | $1.58 \times 10^{-3}$ | $1.55 \times 10^{-3}$ | $1.51 \times 10^{-3}$ | $1.06 \times 10^{-3}$ | $1.06 \times 10^{-3}$ | $1.01 \times 10^{-3}$ |
| 303.15 | $1.68 \times 10^{-3}$ | $1.68 \times 10^{-3}$ | $1.61 \times 10^{-3}$ | $1.45 \times 10^{-3}$ | $1.44 \times 10^{-3}$ | $1.40 \times 10^{-3}$ | $1.97 \times 10^{-3}$ | $1.97 \times 10^{-3}$ | $1.92 \times 10^{-3}$ | $1.40 \times 10^{-3}$ | $1.38 \times 10^{-3}$ | $1.33 \times 10^{-3}$ |
| 308.15 | $2.11 \times 10^{-3}$ | $2.12 \times 10^{-3}$ | $2.07 \times 10^{-3}$ | $1.82 \times 10^{-3}$ | $1.83 \times 10^{-3}$ | $1.81 \times 10^{-3}$ | $2.44 \times 10^{-3}$ | $2.46 \times 10^{-3}$ | $2.43 \times 10^{-3}$ | $1.76 \times 10^{-3}$ | $1.76 \times 10^{-3}$ | $1.72 \times 10^{-3}$ |
| 313.15 | $2.61 \times 10^{-3}$ | $2.61 \times 10^{-3}$ | $2.65 \times 10^{-3}$ | $2.27 \times 10^{-3}$ | $2.29 \times 10^{-3}$ | $2.31 \times 10^{-3}$ | $3.01 \times 10^{-3}$ | $3.02 \times 10^{-3}$ | $3.05 \times 10^{-3}$ | $2.19 \times 10^{-3}$ | $2.20 \times 10^{-3}$ | $2.22 \times 10^{-3}$ |
| 318.15 | $3.17 \times 10^{-3}$ | $3.17 \times 10^{-3}$ | $3.37 \times 10^{-3}$ | $2.83 \times 10^{-3}$ | $2.82 \times 10^{-3}$ | $2.94 \times 10^{-3}$ | $3.69 \times 10^{-3}$ | $3.69 \times 10^{-3}$ | $3.81 \times 10^{-3}$ | $2.69 \times 10^{-3}$ | $2.69 \times 10^{-3}$ | $2.84 \times 10^{-3}$ |
| | **n-Butanol** | | | **Isobutanol** | | | **Ethyl Acetate** | | | **Acetonitrile** | | |
| 278.15 | $3.43 \times 10^{-4}$ | $3.36 \times 10^{-4}$ | $3.69 \times 10^{-4}$ | $4.24 \times 10^{-4}$ | $4.17 \times 10^{-4}$ | $4.47 \times 10^{-4}$ | $7.90 \times 10^{-4}$ | $7.69 \times 10^{-4}$ | $7.68 \times 10^{-4}$ | $1.63 \times 10^{-4}$ | $1.57 \times 10^{-4}$ | $1.69 \times 10^{-4}$ |
| 283.15 | $4.88 \times 10^{-4}$ | $4.89 \times 10^{-4}$ | $5.00 \times 10^{-4}$ | $5.73 \times 10^{-4}$ | $5.88 \times 10^{-4}$ | $5.98 \times 10^{-4}$ | $9.67 \times 10^{-4}$ | $9.70 \times 10^{-4}$ | $9.72 \times 10^{-4}$ | $2.17 \times 10^{-4}$ | $2.18 \times 10^{-4}$ | $2.22 \times 10^{-4}$ |
| 288.15 | $6.72 \times 10^{-4}$ | $6.88 \times 10^{-4}$ | $6.72 \times 10^{-4}$ | $8.07 \times 10^{-4}$ | $8.08 \times 10^{-4}$ | $7.92 \times 10^{-4}$ | $1.20 \times 10^{-3}$ | $1.21 \times 10^{-3}$ | $1.22 \times 10^{-3}$ | $2.87 \times 10^{-4}$ | $2.94 \times 10^{-4}$ | $2.90 \times 10^{-4}$ |
| 293.15 | $9.39 \times 10^{-4}$ | $9.40 \times 10^{-4}$ | $8.93 \times 10^{-4}$ | $1.09 \times 10^{-3}$ | $1.08 \times 10^{-3}$ | $1.04 \times 10^{-3}$ | $1.51 \times 10^{-3}$ | $1.51 \times 10^{-3}$ | $1.52 \times 10^{-3}$ | $3.90 \times 10^{-4}$ | $3.89 \times 10^{-4}$ | $3.75 \times 10^{-4}$ |
| 298.15 | $1.26 \times 10^{-3}$ | $1.25 \times 10^{-3}$ | $1.18 \times 10^{-3}$ | $1.42 \times 10^{-3}$ | $1.42 \times 10^{-3}$ | $1.35 \times 10^{-3}$ | $1.87 \times 10^{-3}$ | $1.88 \times 10^{-3}$ | $1.88 \times 10^{-3}$ | $5.04 \times 10^{-4}$ | $5.02 \times 10^{-4}$ | $4.82 \times 10^{-4}$ |
| 303.15 | $1.62 \times 10^{-3}$ | $1.61 \times 10^{-3}$ | $1.54 \times 10^{-3}$ | $1.82 \times 10^{-3}$ | $1.81 \times 10^{-3}$ | $1.75 \times 10^{-3}$ | $2.32 \times 10^{-3}$ | $2.31 \times 10^{-3}$ | $2.32 \times 10^{-3}$ | $6.37 \times 10^{-4}$ | $6.36 \times 10^{-4}$ | $6.14 \times 10^{-4}$ |
| 308.15 | $2.03 \times 10^{-3}$ | $2.04 \times 10^{-3}$ | $1.99 \times 10^{-3}$ | $2.27 \times 10^{-3}$ | $2.27 \times 10^{-3}$ | $2.24 \times 10^{-3}$ | $2.86 \times 10^{-3}$ | $2.84 \times 10^{-3}$ | $2.84 \times 10^{-3}$ | $7.90 \times 10^{-4}$ | $7.89 \times 10^{-4}$ | $7.76 \times 10^{-4}$ |
| 313.15 | $2.50 \times 10^{-3}$ | $2.51 \times 10^{-3}$ | $2.56 \times 10^{-3}$ | $2.79 \times 10^{-3}$ | $2.80 \times 10^{-3}$ | $2.84 \times 10^{-3}$ | $3.45 \times 10^{-3}$ | $3.46 \times 10^{-3}$ | $3.45 \times 10^{-3}$ | $9.60 \times 10^{-4}$ | $9.63 \times 10^{-4}$ | $9.75 \times 10^{-4}$ |
| 318.15 | $3.03 \times 10^{-3}$ | $3.02 \times 10^{-3}$ | $3.27 \times 10^{-3}$ | $3.39 \times 10^{-3}$ | $3.38 \times 10^{-3}$ | $3.59 \times 10^{-3}$ | $4.20 \times 10^{-3}$ | $4.21 \times 10^{-3}$ | $4.19 \times 10^{-3}$ | $1.15 \times 10^{-3}$ | $1.15 \times 10^{-3}$ | $1.22 \times 10^{-3}$ |
| | **Cyclohexanone** | | | **Acetone** | | | **1,4-Dioxane** | | | **DMF** | | |
| 278.15 | $7.92 \times 10^{-3}$ | $7.99 \times 10^{-3}$ | $7.88 \times 10^{-3}$ | $2.01 \times 10^{-3}$ | $2.03 \times 10^{-3}$ | $2.01 \times 10^{-3}$ | | | | $1.07 \times 10^{-2}$ | $1.08 \times 10^{-2}$ | $1.09 \times 10^{-2}$ |
| 283.15 | $9.41 \times 10^{-3}$ | $9.41 \times 10^{-3}$ | $9.37 \times 10^{-3}$ | $2.39 \times 10^{-3}$ | $2.39 \times 10^{-3}$ | $2.38 \times 10^{-3}$ | | | | $1.45 \times 10^{-2}$ | $1.45 \times 10^{-2}$ | $1.45 \times 10^{-2}$ |
| 288.15 | $1.11 \times 10^{-2}$ | $1.10 \times 10^{-2}$ | $1.11 \times 10^{-2}$ | $2.82 \times 10^{-3}$ | $2.80 \times 10^{-3}$ | $2.81 \times 10^{-3}$ | $6.42 \times 10^{-3}$ | $6.44 \times 10^{-3}$ | $6.38 \times 10^{-3}$ | $1.93 \times 10^{-2}$ | $1.92 \times 10^{-2}$ | $1.91 \times 10^{-2}$ |
| 293.15 | $1.31 \times 10^{-2}$ | $1.30 \times 10^{-2}$ | $1.30 \times 10^{-2}$ | $3.29 \times 10^{-3}$ | $3.28 \times 10^{-3}$ | $3.30 \times 10^{-3}$ | $8.34 \times 10^{-3}$ | $8.27 \times 10^{-3}$ | $8.27 \times 10^{-3}$ | $2.54 \times 10^{-2}$ | $2.51 \times 10^{-2}$ | $2.49 \times 10^{-2}$ |
| 298.15 | $1.52 \times 10^{-2}$ | $1.52 \times 10^{-2}$ | $1.53 \times 10^{-2}$ | $3.84 \times 10^{-3}$ | $3.83 \times 10^{-3}$ | $3.86 \times 10^{-3}$ | $1.06 \times 10^{-2}$ | $1.06 \times 10^{-2}$ | $1.06 \times 10^{-2}$ | $3.24 \times 10^{-2}$ | $3.24 \times 10^{-2}$ | $3.20 \times 10^{-2}$ |
| 303.15 | $1.77 \times 10^{-2}$ | $1.77 \times 10^{-2}$ | $1.79 \times 10^{-2}$ | $4.46 \times 10^{-3}$ | $4.47 \times 10^{-3}$ | $4.49 \times 10^{-3}$ | $1.33 \times 10^{-2}$ | $1.34 \times 10^{-2}$ | $1.35 \times 10^{-2}$ | $4.11 \times 10^{-2}$ | $4.12 \times 10^{-2}$ | $4.09 \times 10^{-2}$ |
| 308.15 | $2.07 \times 10^{-2}$ | $2.07 \times 10^{-2}$ | $2.08 \times 10^{-2}$ | $5.19 \times 10^{-3}$ | $5.21 \times 10^{-3}$ | $5.22 \times 10^{-3}$ | $1.70 \times 10^{-2}$ | $1.70 \times 10^{-2}$ | $1.71 \times 10^{-2}$ | $5.15 \times 10^{-2}$ | $5.18 \times 10^{-2}$ | $5.16 \times 10^{-2}$ |
| 313.15 | $2.41 \times 10^{-2}$ | $2.41 \times 10^{-2}$ | $2.41 \times 10^{-2}$ | $6.07 \times 10^{-3}$ | $6.06 \times 10^{-3}$ | $6.05 \times 10^{-3}$ | $2.15 \times 10^{-2}$ | $2.14 \times 10^{-2}$ | $2.14 \times 10^{-2}$ | $6.45 \times 10^{-2}$ | $6.43 \times 10^{-2}$ | $6.46 \times 10^{-2}$ |
| 318.15 | $2.81 \times 10^{-2}$ | $2.81 \times 10^{-2}$ | $2.78 \times 10^{-2}$ | $7.04 \times 10^{-3}$ | $7.03 \times 10^{-3}$ | $6.99 \times 10^{-3}$ | $2.69 \times 10^{-2}$ | $2.69 \times 10^{-2}$ | $2.67 \times 10^{-2}$ | $7.91 \times 10^{-2}$ | $7.91 \times 10^{-2}$ | $8.01 \times 10^{-2}$ |

[a] Standard uncertainties $u$ are $u(T)$ =0.02 K, $u(p)$ = 400 Pa, the relative standard uncertainty of mole solubility $u_r$ is $u_r(x)$ = 0.06.

It can be easily found that the mole fraction solubility of ethyl candesartan increases with the rising temperature in all solvents. The solubility values of ethyl candesartan were found to be maximum in DMF ($7.91 \times 10^{-2}$), followed by cyclohexanone ($2.81 \times 10^{-2}$), 1,4-dioxanone ($2.688 \times 10^{-2}$), acetone ($7.04 \times 10^{-3}$), ethyl acetate ($4.20 \times 10^{-3}$), n-propanol ($3.69 \times 10^{-3}$), isobutanol ($3.38 \times 10^{-3}$), methanol ($3.17 \times 10^{-3}$), n-butanol ($3.03 \times 10^{-3}$), ethanol ($2.83 \times 10^{-3}$), isopropanol ($2.69 \times 10^{-3}$), and acetonitrile ($1.15 \times 10^{-2}$) at a temperature of 318.15 K. Similar results of solubility sequence from large to small were also obtained in other temperatures.

In alcohols, there is no obvious regularity in the order of molar fraction solubility. However, the mass fraction solubility shows a certain degree of trend at 318.15 K; the maximum data was observed in methanol, and the minimum was found in n-butanol. As can be seen from Table 2, there is little difference in the solubility of mole fractions in alcohols, and the trend of solubility curve is affected by the molecular weight of the solvent. The order of mole fraction solubility values in non-alcoholic solvents, from large to small, is DMF > cyclohexanone > 1,4-dioxanone > acetone > ethyl acetate > acetonitrile. Through the analysis of solvent properties in non-protonic select solvents, it was found that the order of solubility is consistent with the sequence of hydrogen bond basicity ($\beta$) with the exception of 1,4-dioxane and acetone ($\beta_{DMF}$ = 0.69, $\beta_{cyclohexanone}$ = 0.53, $\beta_{1,4\text{-dioxane}}$ = 0.37, $\beta_{acetone}$ = 0.43, $\beta_{ethyl\ acetate}$ = 0.45 and $\beta_{acetonitrile}$ = 0.40). This phenomenon could also be observed with the values of dipolarity/polarizability ($\pi^*$) except in 1,4-dioxane and acetonitrile ($\pi^*_{DMF}$ = 0.88, $\pi^*_{cyclohexanone}$ = 0.76, $\pi^*_{1,4\text{-dioxane}}$ = 0.55, $\pi^*_{acetone}$ = 0.71, $\pi^*_{ethyl\ acetate}$ = 0.55 and $\pi^*_{acetonitrile}$ = 0.75) [20]. In aprotic solvents, except for individual solvents, the solubility data decreases with the decreasing values of $\beta$ and $\pi^*$, which indicates that the dissolution process of ethyl candesartan in selected pure solvents is complicated, which may be caused by some combination of multiple factors. By ana-

lyzing the molecular structure of the solute, it can be found that the -NH structure on the heterocycle, as the only hydrogen proton donor, forms a hydrogen bond with the solvent molecule. Especially in polar aprotic solvents, solute molecules play the role of Lewis acid.

### 3.3. Correlation Section

Based on the non-ideal solution, the modified Apelblat equation, Equation (1), has already been one of the most commonly and widely used models in solubility correlation, especially in engineering applications. It has a high accuracy to describe the function between the solubility data and temperature in Kelvin, which can be expressed as follows [21–25]:

$$\ln x_{w,T} = A + \frac{B}{T} + C \ln T \tag{1}$$

where $x_{w,T}$ is the mole fraction solubility of ethyl candesartan in different solvents at temperature $T$ in Kelvin. $A$, $B$, and $C$ refer to the equation parameters.

The $\lambda h$ equation is another semi-empirical equation, which can also provide a good description of the solid–liquid equilibrium in different solvents, as presented in Equation (2) [26–28]:

$$\ln\left[1 + \frac{\lambda(1-x)}{x}\right] = \lambda h\left(\frac{1}{T} - \frac{1}{T_m}\right) \tag{2}$$

where $\lambda$ and $h$ are equation parameters, and $T_m$ is the melting temperature of ethyl candesartan, 432.15 K, cited from ref. [29].

In order to evaluate the fitting accuracy and applicability of the selected two models and for ethyl candesartan, the average relative deviation (*ARD*) was proposed to compare the correlation results and is shown in Equation (3):

$$ARD = \frac{1}{N}\sum_{i=1}^{N}\left|\frac{x_i^e - x_i^c}{x_i^e}\right| \tag{3}$$

In Equation (3), $N$ is the number of experimental points in each solvent. $x_i^e$ and $x_i^c$ refer to the experimental and calculated mole fraction solubility values. The values of *ARD* along with model parameters are listed in Table 3. All values of *ARD* in the modified Apelblat equation are smaller than that in the $\lambda h$ equation. Moreover, the largest *ARD* data in the modified Apelblat equation is 1.9% and observed in isopropanol; the maximum data in the $\lambda h$ equation is 4.3% and found in *n*-butanol. The results may indicate two selected models can provide a satisfactory correlation solubility of ethyl candesartan as crucial data and model parameters in the industrial production process, while the modified Apelblat equation shows the more suitable correlation with experimental data of ethyl candesartan in selected pure solvents at all investigated temperatures.

**Table 3.** The results of model parameters along with *ARD* values.

| Solvent | Modified Apelblat Equation | | | | $\lambda h$ Equation | | |
|---|---|---|---|---|---|---|---|
| | *A* | *B* | *C* | $10^2$ *ARD* | 100 $\lambda$ | *h* | $10^2$ *ARD* |
| Methanol | 358.1 | −20,259.5 | −52.1 | 1.1 | 16.2 | 28,930.2 | 3.6 |
| Ethanol | 226.9 | −14,430.3 | −32.5 | 0.7 | 14.5 | 32,637.8 | 2.5 |
| *n*-Propanol | 193.6 | −12,615.4 | −27.7 | 1.3 | 13.5 | 32,011.2 | 2.5 |
| Isopropanol | 340.4 | −19,633.8 | −49. 4 | 1.9 | 15.5 | 31,305.1 | 2.9 |
| *n*-Butanol | 428.4 | −23,457.7 | −62.6 | 0.8 | 17.2 | 27,923.4 | 4.3 |
| Isobutanol | 320.3 | −18,454.9 | −46.5 | 0.6 | 15.6 | 29,266.2 | 3.8 |
| Ethyl Acetate | −37.3 | −1822.6 | 6.5 | 0.7 | 8.4 | 43,701.2 | 0.9 |
| Acetonitrile | 295.4 | −17,202.8 | −43.0 | 0.9 | 4.3 | 101,448.0 | 3.1 |
| Cyclohexanone | −78.8 | 945.9 | 12.5 | 0.3 | 22.9 | 11,560.7 | 0.5 |
| Acetone | −79.8 | 958.6 | 12.5 | 0.4 | 5.2 | 49,105.4 | 0.5 |
| 1,4-Dioxane | −75.2 | −522.7 | 12.7 | 0.5 | 100.5 | 4352.1 | 0.7 |
| DMF | 113.4 | −8933.7 | −15.2 | 0.5 | 359.8 | 1255.3 | 1.0 |

### 3.4. Quantitative Analysis of Interactions

The solute–solvent interactions are important parameters for the estimation of the solubility of the solute in a given solvent system. In this work, different parameters were used to quantify the solute–solvent interactions that occurred in the dissolution process, including Abraham solvation parameters ($AP_i$), Hansen solubility parameters ($HP_i$), and Catalan parameters ($CP_i$). [30,31] The numerical values of $AP_i$, $HP_i$, and $CP_i$ for the investigated solvents were tabulated in Table 4 [30–33]. The combined model could be presented as:

$$\ln x = \left( \alpha_0 + \sum_{i=1}^{5} \alpha_{i,Ab} AP_i + \sum_{i=1}^{3} \alpha_{i,HP} HP_i + \sum_{i=1}^{4} \alpha_{i,CP} CP_i \right) + \left( \frac{\beta_0 + \sum_{i=1}^{5} \beta_{i,Ab} AP_i + \sum_{i=1}^{3} \beta_{i,HP} HP_i + \sum_{i=1}^{4} \beta_{i,CP} CP_i}{T} \right) \quad (4)$$

where $\alpha$ and $\beta$ terms are the model parameters computed using regression analysis. Equation (4) was obtained from combining the van't Hoff model and the $AP_i$, $HP_i$, and $CP_i$ parameters. The significant ($p < 0.05$) variables which are obtained from the regression analysis of solubility data in mono solvents at various temperatures is:

$$\ln x = (-23.416(\pm 4.851) + 6.22(\pm 1.075)v + 0.528(\pm 0.062)\delta_h - 2.02(\pm 0.345)SA) +$$
$$\left( \frac{5111.558(\pm 1460.518) - 662.582(\pm 69.119)c + 279.251(\pm 13.462)s - 128.103(\pm 6.114)b - 2514.547(\pm 325.425)v - 142.667(\pm 18.05)\delta_h + 3594.333(\pm 224.505)SP}{T} \right) \quad (5)$$

where $c$, $s$, and $v$ are $AP_i$ parameters, $\delta_h$ is the $HP_i$ parameter, and $SA$ and $SP$ are $CP_i$ parameters. The quantitative analysis of solvent–solute interactions is presented in Equation (5) with $R = 0.995$ ($N = 106$). The resulted $ARD$ values for back-calculated solubility data using Equation (5) are listed in Table 5; moreover, the predicted values are presented as well. The maximum (20.6%) and minimum (3.3%) $ARD$ values were observed for $n$-propanol and isopropanol data sets. The reasons for large ARD data could be related to the error in the experimental solubility determinations, inaccurate values of $AP_i$, $HP_i$, and $CP_i$ parameters, and some other undefined errors. In addition, the proposed model possesses some weakness in cross validation, since it employs lots of model parameters; however, it is a starting point to model the solubility data in mono solvents at various temperatures using a single linear model.

**Table 4.** The numerical values of Abraham solvation parameters ($AP_i$), Hansen solubility parameters ($HP_i$), and Catalan parameters [a].

| Solvent | Abraham | | | | | | Hansen | | | Catalan | | | |
|---|---|---|---|---|---|---|---|---|---|---|---|---|---|
| | $c$ | $e$ | $s$ | $a$ | $b$ | $v$ | $\delta_d$ | $\delta_p$ | $\delta_h$ | $SP$ | $SdP$ | $SA$ | $SB$ |
| Methanol | 0.28 | 0.33 | −0.71 | 0.24 | −3.32 | 3.55 | 15.10 | 12.30 | 22.3 | 0.61 | 0.9 | 0.61 | 0.55 |
| Ethanol | 0.22 | 0.47 | −1.04 | 0.33 | −3.6 | 3.86 | 15.75 | 8.90 | 19.61 | 0.64 | 0.78 | 0.4 | 0.66 |
| $n$-Propanol | 0.13 | 0.38 | −0.92 | 0.42 | −3.49 | 3.82 | 16.00 | 6.80 | 17.40 | 0.66 | 0.75 | 0.37 | 0.78 |
| Isopropanol | 0.10 | 0.34 | −1.05 | 0.41 | −3.83 | 4.03 | 15.8 | 6.10 | 16.40 | 0.63 | 0.81 | 0.28 | 0.83 |
| $n$-Butanol | 0.17 | 0.40 | −1.01 | 0.06 | −3.96 | 4.04 | 16.00 | 5.70 | 15.80 | 0.67 | 0.66 | 0.34 | 0.81 |
| Isobutanol | 0.19 | 0.35 | −1.13 | 0.02 | −3.57 | 3.97 | 15.80 | 5.70 | 14.50 | 0.66 | 0.71 | 0.22 | 0.89 |
| Ethyl Acetate | 0.33 | 0.37 | −0.45 | −0.70 | −4.90 | 4.15 | 15.80 | 5.30 | 7.20 | 0.66 | 0.60 | 0.00 | 0.54 |
| Acetonitrile | 0.41 | 0.08 | 0.33 | −1.57 | 4.39 | 3.36 | 11.59 | 12.95 | 16.34 | 0.65 | 0.97 | 0.04 | 0.29 |
| Cyclohexanone | 0.04 | 0.23 | 0.06 | −0.98 | −4.84 | 4.32 | 17.80 | 6.30 | 5.10 | 0.77 | 0.75 | 0.00 | 0.48 |
| Acetone | 0.31 | 0.31 | −0.12 | −0.61 | −4.75 | 3.94 | 15.50 | 10.40 | 7.00 | 0.65 | 0.91 | 0.00 | 0.48 |
| 1,4-Dioxane | 0.10 | 0.35 | −0.08 | −0.56 | −4.83 | 4.17 | 19.00 | 1.80 | 7.40 | 0.74 | 0.31 | 0.00 | 0.44 |
| N,N-Dimethylformamide | −0.31 | −0.06 | 0.34 | 0.36 | −4.87 | 4.49 | 17.4 | 13.70 | 11.3 | 0.76 | 0.98 | 0.03 | 0.61 |

[a] cited from Refs. [30–33].

**Table 5.** The values of back-calculated logarithmic solubility data using Equation (5) along with *ARD* results.

| T | | | | | Solvent | | | |
|---|---|---|---|---|---|---|---|---|
| | ln$x$ | ln$x$ (Pred) | ln$x$ | ln$x$ (Pred) | ln$x$ | ln$x$ (Pred) | ln$x$ | ln$x$ (Pred) |
| | Methanol | | Ethanol | | *n*-Propanol | | Isopropanol | |
| 278.15 | −7.87 | −7.91 | −8.03 | −8.08 | −7.57 | −7.39 | −8.08 | −8.11 |
| 283.15 | −7.55 | −7.61 | −7.67 | −7.75 | −7.31 | −7.11 | −7.78 | −7.79 |
| 288.15 | −7.26 | −7.32 | −7.38 | −7.44 | −7.01 | −6.83 | −7.49 | −7.49 |
| 293.15 | −6.93 | −7.04 | −7.09 | −7.14 | −6.71 | −6.56 | −7.16 | −7.19 |
| 298.15 | −6.63 | −6.76 | −6.8 | −6.85 | −6.45 | −6.31 | −6.85 | −6.91 |
| 303.15 | −6.39 | −6.50 | −6.53 | −6.57 | −6.23 | −6.06 | −6.57 | −6.64 |
| 308.15 | −6.16 | −6.25 | −6.31 | −6.3 | −6.01 | −5.82 | −6.34 | −6.37 |
| 313.15 | −5.95 | −6.00 | −6.09 | −6.04 | −5.81 | −5.58 | −6.12 | −6.11 |
| 318.15 | −5.75 | −5.76 | −5.87 | −5.78 | −5.6 | −5.36 | −5.92 | −5.86 |
| *ARD* | 7.0% | | 5.3% | | 20.6% | | 3.3% | |
| | *n*-Butanol | | Isobutanol | | Ethyl Acetate | | Acetonitrile | |
| 278.15 | −7.98 | −7.82 | −7.76 | −7.87 | −7.14 | −7.09 | −8.72 | −8.62 |
| 283.15 | −7.62 | −7.51 | −7.46 | −7.58 | −6.94 | −6.85 | −8.44 | −8.36 |
| 288.15 | −7.31 | −7.22 | −7.12 | −7.31 | −6.73 | −6.62 | −8.16 | −8.11 |
| 293.15 | −6.97 | −6.94 | −6.82 | −7.04 | −6.5 | −6.41 | −7.85 | −7.87 |
| 298.15 | −6.68 | −6.66 | −6.56 | −6.78 | −6.28 | −6.19 | −7.59 | −7.63 |
| 303.15 | −6.42 | −6.4 | −6.31 | −6.52 | −6.07 | −5.99 | −7.36 | −7.41 |
| 308.15 | −6.2 | −6.14 | −6.09 | −6.28 | −5.86 | −5.79 | −7.14 | −7.19 |
| 313.15 | −5.99 | −5.9 | −5.88 | −6.04 | −5.67 | −5.6 | −6.95 | −6.98 |
| 318.15 | −5.8 | −5.66 | −5.69 | −5.82 | −5.47 | −5.42 | −6.76 | −6.77 |
| *ARD* | 8.4% | | 15.7% | | 8.1% | | 4.8% | |
| | Cyclohexanone | | Acetone | | 1,4-Dioxane | | *N,N*-Dimethylformamide | |
| 278.15 | −4.84 | −5 | −6.21 | −6.32 | | | −4.54 | −4.45 |
| 283.15 | −4.67 | −4.8 | −6.04 | −6.12 | | | −4.23 | −4.18 |
| 288.15 | −4.5 | −4.61 | −5.87 | −5.93 | −5.05 | −4.81 | −3.95 | −3.93 |
| 293.15 | −4.34 | −4.43 | −5.72 | −5.75 | −4.79 | −4.62 | −3.67 | −3.69 |
| 298.15 | −4.18 | −4.25 | −5.56 | −5.57 | −4.55 | −4.44 | −3.43 | −3.45 |
| 303.15 | −4.04 | −4.08 | −5.41 | −5.4 | −4.32 | −4.26 | −3.19 | −3.22 |
| 308.15 | −3.88 | −3.91 | −5.26 | −5.23 | −4.07 | −4.08 | −2.97 | −3 |
| 313.15 | −3.72 | −3.75 | −5.1 | −5.07 | −3.84 | −3.92 | −2.74 | −2.79 |
| 318.15 | −3.57 | −3.6 | −4.96 | −4.92 | −3.62 | −3.75 | −2.54 | −2.58 |
| *ARD* | 7.3% | | 4.3% | | 12.0% | | 3.9% | |

## 4. Conclusions

The mole fraction solubility of ethyl candesartan in selected mono solvents within the temperature range of 278.15 to 318.15 K was measured. The largest solubility data of ethyl candesartan were found in DMF, followed by cyclohexanone, 1,4-dioxanone, acetone, ethyl acetate, *n*-propanol, isobutanol, methanol, *n*-butanol, ethanol, isopropanol, and acetonitrile at each temperature. In aprotic solvents, except for individual solvents, the solubility data decreases with the decreasing values of hydrogen bond basicity ($\beta$) and dipolarity/polarizability ($\pi^*$). The results of statistical analysis show that the modified Apelblat equation is the more suitable correlation of experimental data for ethyl candesartan in selected mono solvents at all investigated temperatures. Moreover, the results may indicate the selected two models can provide satisfactory correlation solubility of ethyl candesartan as crucial data and model parameters in the industrial production process

**Funding:** This research received no external funding.

**Data Availability Statement:** Data are contained within the article.

**Acknowledgments:** This paper is in memory of the first anniversary of the death of my grandfather (Fangshan Du). As time flies, you have been away for a year. I miss you very much and wish you all the best in heaven.

**Conflicts of Interest:** The author declares no conflict of interest.

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
