# Peer review of "The Solubility of Ethyl Candesartan in Mono Solvents and Investigation of Intermolecular Interactions"

_liquids, doi:10.3390/liquids2040023_

Round 1
Reviewer 1 Report
The author reports the solubility of ethylcandesartan in twelve pure solvents at 9 temperatures. The solubility process of prodrug is endothermic. The experimental solubility was correlated with two models and an analysis of the interactions was performed.
The paper is well written, the analytical technique to determine the solubility of Ethyl Candesartan (HPLC) guarantees the quality of the experimental data. By recalculating some data, the accuracy and quality of the data can be verified.
X-ray diffraction analysis allows us to ensure that the solubility changes as a function of solubility are a consequence of this factor and not due to changes such as polymorphism and/or solvate formation.
In this context, I advise to accept the manuscript, after some minor corrections.
1. line 117: report solubility units (mole fraction)
2. Unify how solubility is expressed: x10^n or e^n throughout the paper. I recommend x10^n
3. Table 3. Unify the number of decimal places, I would think that one is enough taking into account the value of the parameters
4. Table 4. Unify the number of decimal places, and report the article or book from which the values were taken. In references 27 and 28 there are no reports of these data.
5. Table 5. Indicate that the solubility reported is the logarithmic solubility.
ble 2:
Author Response
The author reports the solubility of ethylcandesartan in twelve pure solvents at 9 temperatures. The solubility process of prodrug is endothermic. The experimental solubility was correlated with two models and an analysis of the interactions was performed.
The paper is well written, the analytical technique to determine the solubility of Ethyl Candesartan (HPLC) guarantees the quality of the experimental data. By recalculating some data, the accuracy and quality of the data can be verified.
X-ray diffraction analysis allows us to ensure that the solubility changes as a function of solubility are a consequence of this factor and not due to changes such as polymorphism and/or solvate formation.
In this context, I advise to accept the manuscript, after some minor corrections.
- line 117: report solubility units (mole fraction)
It was corrected in the manuscript.
- Unify how solubility is expressed: x10^n or e^n throughout the paper. I recommend x10^n
It was revised and solubility was expressed as x10^n in Table 2.
- Table 3. Unify the number of decimal places, I would think that one is enough taking into account the value of the parameters
It was revised in the manuscript.
- Table 4. Unify the number of decimal places, and report the article or book from which the values were taken. In references 27 and 28 there are no reports of these data.
It was corrected in the manuscript. Model parameters were cited from the supporting information and body part in Refs 27 and 28.
- Table 5. Indicate that the solubility reported is the logarithmic solubility.
It was corrected in Table 5. We are sorry for our carelessness.
Reviewer 2 Report
The authors have reported new solubility data of ethyl candesartan in several mono solvents at different temperatures. The measured solubility data of investigated drug was corrected with some computational models and various solubility parameters are reported. Overall, the work is interesting and will be beneficial for the fellow researchers. However, it requires extensive revision before its publication. My comments are as follows:
Abstract: The quantitative results regarding the computational models are missing in the abstract. Authors are advised to include some quantitative results of computational models for the better understanding of the models.
Line 41: It should be Figure 1.
Figure 1: Kindly remove the color from Figure 1.
Introduction: The introduction is quite shallow. The solubility part is too poor. Authors are advised to add recent literature about the importance of solubility and different solubility approaches. You can consult the following articles to make this manuscript more useful to the readers.
J. Mol. Liq. 299: E112211 (2020); J. Mol. Liq. 307: E112970 (2020); ACS Omega 5: 1708-1716 (2020); J. Therm. Anal. Calorim. 142: 1437-1446 (2020); Molecules 25: E1559 (2020); J. Mol. Liq. 324: E115146 (2021); J. Mol. Liq. 331: E115700 (2021); J. Mol. Liq. 340: E117268 (2021); Alexandria Eng. J. 61: 9119-9128 (2022); J. Mol. Liq. 348: E118057 (2022).
Data presentation: Authors are advised to present data either in Table or Figure. Please avoid the duplication in order to save the space of the journal.
Please provide error graphs in the figure; where are required.
Please compare your results with previous studies and mention clearly how your work is important in comparison to already been reported.
Avoid abbreviations before giving their explanation.
Results and discussion: Authors are advised to include the main limitation of work at the end of results and discussion section and just before the conclusion.
Discussion: The authors have presented the results in most of the part. The discussion part is too poor. Please improve the discussion of results.
Conclusion: The conclusion should be concise and to the point indicating the application of the work.
Several subsections, such as Funding, Acknowledgement, Ethical approval, and Conflict of interests etc. are missing. Kindly include them.
Author Response
The authors have reported new solubility data of ethyl candesartan in several mono solvents at different temperatures. The measured solubility data of investigated drug was corrected with some computational models and various solubility parameters are reported. Overall, the work is interesting and will be beneficial for the fellow researchers. However, it requires extensive revision before its publication. My comments are as follows:
Abstract: The quantitative results regarding the computational models are missing in the abstract. Authors are advised to include some quantitative results of computational models for the better understanding of the models.
Some quantitative results of computational models were added in the Abstract.
Line 41: It should be Figure 1.
It was corrected in the manuscript.
Figure 1: Kindly remove the color from Figure 1.
It was corrected in Figure 1.
Introduction: The introduction is quite shallow. The solubility part is too poor. Authors are advised to add recent literature about the importance of solubility and different solubility approaches. You can consult the following articles to make this manuscript more useful to the readers. J. Mol. Liq. 299: E112211 (2020); J. Mol. Liq. 307: E112970 (2020); ACS Omega 5: 1708-1716 (2020); J. Therm. Anal. Calorim. 142: 1437-1446 (2020); Molecules 25: E1559 (2020); J. Mol. Liq. 324: E115146 (2021); J. Mol. Liq. 331: E115700 (2021); J. Mol. Liq. 340: E117268 (2021); Alexandria Eng. J. 61: 9119-9128 (2022); J. Mol. Liq. 348: E118057 (2022).
Some sentences were added into the introduction, and some references provided are cited. Thanks for your kind reminding.
Data presentation: Authors are advised to present data either in Table or Figure. Please avoid the duplication in order to save the space of the journal.
In order to save the space of the journal, the figure was deleted in the revised version.
Please provide error graphs in the figure; where are required.
Because the solubility difference in alcohol solvent is small, and the figure with error bar has aesthetic defects, I use the relative standard uncertainty to measure the reliability of the experimental value. Thanks for your good advice.
Please compare your results with previous studies and mention clearly how your work is important in comparison to already been reported.
I did my best to consult a large number of literatures and found no reports on the solubility of solute compound.
Avoid abbreviations before giving their explanation.
It was corrected in the manuscript.
Results and discussion: Authors are advised to include the main limitation of work at the end of results and discussion section and just before the conclusion.
The sentence of “The proposed model possesses some weakness in cross-validation, since it employs lots of model parameter, however, it is a starting point to model the solubility data in mono-solvents at various temperatures using a single linear model.” was provided in the revised version.
Discussion: The authors have presented the results in most of the part. The discussion part is too poor. Please improve the discussion of results.
Some sentences including “By analyzing the molecular structure of the solute, it can be found that the -NH structure on the heterocycle, as the only hydrogen proton donor, forms a hydrogen bond with the solvent molecule. Especially in polar aprotic solvents, solute molecules play the role of Lewis acid.” were added into the revised version.
Conclusion: The conclusion should be concise and to the point indicating the application of the work.
It was corrected in the manuscript.
Several subsections, such as Funding, Acknowledgement, Ethical approval, and Conflict of interests etc. are missing. Kindly include them.
It was added in the revised version. Thanks for your kind reminding.
Round 2
Reviewer 2 Report
The authors have addressed the previous concerns. The manuscript is suitable for publication in its present form.